# Bio-fabrication of stem-cell-incorporated corneal epithelial and stromal equivalents from silk fibroin and gelatin-based biomaterial for canine corneal regeneration

Chutirat Torsahakul[1,2,3], Nipan Israsena[4,5], Supaporn Khramchantuk[6], Juthamas Ratanavaraporn[7,8,9], Sirakarnt Dhitavat[10], Watchareewan Rodprasert[2,3], Sirirat Nantavisai[2,3,11], Chenphop Sawangmake[2,3,12,13] *

1 Graduate program in Veterinary Bioscience, Faculty of Veterinary Science, Chulalongkorn University, Bangkok, Thailand, 2 Veterinary Stem Cell and Bioengineering Research Unit, Faculty of Veterinary Science, Chulalongkorn University, Bangkok, Thailand, 3 Veterinary Stem Cell and Bioengineering Innovation Center (VSCBIC), Faculty of Veterinary Science, Chulalongkorn University, Bangkok, Thailand, 4 Stem Cell and Cell Therapy Research Unit, Faculty of Medicine, Chulalongkorn University, Bangkok, Thailand, 5 Department of Pharmacology, Faculty of Medicine, Chulalongkorn University, Bangkok, Thailand, 6 Excellence Center for Stem Cell and Cell Therapy, King Chulalongkorn Memorial Hospital, The Thai Red Cross Society, Bangkok, Thailand, 7 Biomedical Engineering Program, Faculty of Engineering, Chulalongkorn University, Bangkok, Thailand, 8 Biomedical Engineering Research Center, Faculty of Engineering, Chulalongkorn University, Bangkok, Thailand, 9 Biomedical Engineering for Medical and Health Research Unit, Faculty of Engineering, Chulalongkorn University, Bangkok, Thailand, 10 Biochemistry Unit, Department of Physiology, Faculty of Veterinary Science, Chulalongkorn University, Bangkok, Thailand, 11 Academic Affairs, Faculty of Veterinary Science, Chulalongkorn University, Bangkok, Thailand, 12 Department of Pharmacology, Faculty of Veterinary Science, Chulalongkorn University, Bangkok, Thailand, 13 Center of Excellence for Regenerative Dentistry (CERD), Faculty of Dentistry, Chulalongkorn University, Bangkok, Thailand

* chenphop.s@chula.ac.th, chenphop@gmail.com

## Abstract

Corneal grafts are the imperative clinical treatment for canine corneal blindness. To serve the growing demand, this study aimed to generate tissue-engineered canine cornea in part of the corneal epithelium and underlying stroma based on canine limbal epithelial stem cells (cLESCs) seeded silk fibroin/gelatin (SF/G) film and canine corneal stromal stem cells (cCSSCs) seeded SF/G scaffold, respectively. Both cell types were successfully isolated by collagenase I. SF/G corneal films and stromal scaffolds served as the prospective substrates for cLESCs and cCSSCs by promoting cell adhesion, cell viability, and cell proliferation. The results revealed the upregulation of *tumor protein P63* (*P63*) and *ATP-binding cassette super-family G member 2* (*Abcg2*) of cLESCs as well as *Keratocan* (*Kera*), *Lumican* (*Lum*), *aldehyde dehydrogenase 3 family member A1* (*Aldh3a1*) and *Aquaporin 1* (*Aqp1*) of differentiated keratocytes. Moreover, immunohistochemistry illustrated the positive staining of tumor protein P63 (P63), aldehyde dehydrogenase 3 family member A1 (Aldh3a1), lumican (Lum) and collagen I (Col-I), which are considerable for native cornea. This study manifested a feasible platform to construct tissue-engineered canine cornea for functional grafts and positively contributed to the body of knowledge related to canine corneal stem cells.

**Data Availability Statement:** All relevant data are within the manuscript and its Supporting Information files.

**Funding:** CT was supported by the 100th Anniversary Chulalongkorn University Fund for Doctoral Scholarship; the 90th Anniversary Chulalongkorn University Fund; and Veterinary Stem Cell and Bioengineering Research Unit, Ratchadaphiseksomphot Endowment Fund of Chulalongkorn University. JR was supported by Chulalongkorn University: CU_GR_62_98_21_15. CS was supported by research supporting grant of the Faculty of Veterinary Science; Chulalongkorn Academic Advancement into Its 2nd Century Project; Center of Excellence for Regenerative Dentistry(CERD), Faculty of Dentistry, Chulalongkorn University; Veterinary Stem Cell and Bioengineering Research Unit, Ratchadaphiseksomphot Endowment Fund, Chulalongkorn University; and Government Research Fund. The funders had no role in study design, data collection and analysis, decision to publish, or preparation of manuscript.

**Competing interests:** The authors have declared that no competing interests exist.

# Introduction

The cornea is a transparent soft tissue located at the outermost layer of the eye. In humans and animals, corneas consist of 5 recognized layers, including 3 cellular layers (epithelium, stroma, and endothelium) and 2 acellular layers (Bowman's layer and Descemet's membrane) [1]. However, Bowman's membrane is not found in carnivores, as it is substituted by special stroma condensation [2].

Corneal blindness is the third most common cause of vision loss, reported in approximately 4.9 million cases for bilateral corneal blindness worldwide [3]. Among these cases in humans, trauma resulting in corneal ulcer, keratitis, and trachoma are the major causes [4]. Likewise, canines also suffer from corneal blindness. The prevalence of canine ulcerative disease has risen to 0.80% in England (834 from 104,233 dogs) [5]. Corneal grafts have become the most common treatment in canines, including conjunctival autograft, nictitating membrane flap [6], human amniotic membrane [7], equine amniotic membrane [8], canine amniotic membrane [9], and porcine urinary bladder submucosa (ACell Vet$^{TM}$) [10]. However, several critical disadvantages of corneal grafts remain, such as disease transmission, contamination, limited tissue shelf life, specific storage condition (−86°C) and graft rejection associated with biological variability [7, 11].

Recently, to rectify those problems, alternative biosynthetic corneal substitutes have been synthesized with cell culturing and tissue engineering. Corneal stem-cell-based therapy offers promising cell sources in tissue engineering, such as limbal epithelial stem cells (LESCs) and corneal stromal stem cells (CSSCs) [12]. LESCs and CSSCs have stem cell properties, e.g. clonal growth *in vitro*, extended lifespan, and the ability of differentiation into corneal cells [13]. Hence, LESCs and CSSCs are worth investigating for isolation, characterization, culture, and differentiation in a tissue-engineered scaffold.

Various materials have been used to fabricate three-dimensional (3-D) biocompatible scaffolds, including natural and synthetic polymers [14]. Natural polymers often have superior biocompatibility, for example, collagen, silk fibroin, gelatin, and chitosan. Consider silk fibroin (SF): SF has demonstrated the advantages of being easily fabricated into various forms, having good tensile strength, high availability, high degradability, transparency, and a non-immunogenic response [15]. Generally, silk composites of 2 main proteins: water-soluble sericin and water-insoluble fibroin. SF is derived by degumming and dissolving with $Na_2CO_3$ and LiBr, respectively [16]. The aforementioned advantages and its natural optical clarity make it a favorable replacement for ocular tissue. There are several supported reports with regard to corneal epithelial scaffold [17], corneal stromal scaffold [18], and co-culturing systems *in vitro* [17, 19].

Nonetheless, some disadvantages have been found; SF has slow degradation rate and a yellow-tinted color [20–22]. To increase SF's degradation rate, it can be incorporated with other rapid-degrading materials such as gelatin [23]. Gelatin is derived from partial hydrolysis of collagen, and it has indicated superb biocompatibility and biodegradability. Regarding corneal bioengineering, gelatin has been used for endothelial cell sheet grafts *in vivo* with satisfying outcomes [24].

This study aimed to investigate silk modified with gelatin film and scaffold for culturing canine limbal epithelial stem cells (cLESCs) and canine corneal stromal stem cells (cCSSCs) to fabricate 3-D synthetic corneal tissue. The outcome of this study was a functional, transplantable corneal graft for canine corneal defect. In addition, isolation, and culture techniques of cLESCs and cCSSCs contribute to the body of knowledge on cell culture in veterinary medicine and are worth further experimentation.

## Materials and methods

### Preparation of silk fibroin solution (SF)

*B. mori* Thai silkworm "hybrid silkworm (J108 X Nanglai strain)" cocoons were gratefully received from the Queen Sirikit Sericulture Center, Sisaket province, Thailand. To prepare the silk solution, silk cocoons were degummed by boiling for 30 min in a 0.02 M $Na_2CO_3$ solution (Ajax Finechem, Australia). The silk fibers were then rinsed with 5 liters of deionized water 4–5 times and dried overnight. After that, they were dissolved in a 9.3 M LiBr solution (Sigma-Aldrich Corporation, USA) at 60˚C for 4 h and dialyzed through a dialysis bag (MWCO 12000–16000, Vikase Company Inc., Osaka, Japan) against distilled water (DI) for 3 days. Impurities were removed by centrifugation at 9,000 rpm (approximately 12,700 g) at 4˚C for 20 min, twice. The obtained SF solution ranged from 6–7% w/w [25, 26].

### Preparation of gelatin (G)

Type A gelatin produced from porcine skin collagen via acidic treatment (Nitta Gelatin Co., Japan; isoelectric point, pl~7–9; MW = 100,616) was provided. 5% w/w gelatin was prepared with 5 g gelatin A and 100 g of DI. The solution was mixed by stirring at 40˚C until homogeneous.

### Preparation of silk fibroin/gelatin (SF/G) corneal films

A silk fibroin/gelatin (SF/G) solution at a ratio of 30/70 and with a final solid concentration of 5% w/w was prepared by gently stirring for 10 min at room temperature (RT). Then, 2 mL of mixed solution was cast in a 5x5 cm squamous polystyrene tray and dried in a laminar hood overnight to obtain the SF/G corneal films. The films were cut into circular shapes with 1 cm diameters by a sterile biopsy punch (Miltex, USA) and annealed in water-filled desiccators under vacuum for 24 h. The films were sterilized by gas plasma before use.

### Preparation of silk fibroin/gelatin (SF/G) stromal scaffolds

15 mL of the SF/G solution was prepared and cast on a 5x5 cm squamous polystyrene tray. The pores were created by freezing at -80˚C overnight before lyophilization at -70˚C for 48 h. After that, the porous scaffolds were crosslinked with a dehydrothermal technique at 140˚C for 72 h under vacuum in a vacuum oven. The air interphase area was removed using a micro-tome blade to remove the upper part of the scaffolds with less porosity prior to cutting them into circular shapes with a 1-cm-diameter sterile biopsy punch (Miltex, USA). The 1 mm thickness of the scaffold was then confirmed with a Vernier Caliper and slid using microtome blade. The scaffolds were weighed for estimating their density. The scaffolds with a similar weight were selected and sterilized by gas plasma before use.

### Material morphological and structural characterization

To assess SF/G corneal films, SF/G stromal scaffolds, cLESCs seeded SF/G corneal films, cCSSCs seeded SF/G stromal scaffolds, and bio-fabricated cornea, scanning electron microscopy (SEM) was used to analyze pore diameter, surface, thickness, cell morphology, and overall structure. Dried samples (SF/G corneal films, SF/G stromal scaffolds) were gold-sputtered before SEM imaging. The thickness of materials and pore size were measured by ImageJ (US National Institute of Health, NIH). Meanwhile, wet samples (cLESCs seeded SF/G corneal films, cCSSCs seeded SF/G stromal scaffolds, and bio-fabricated cornea) were fixed with 2.5% glutaraldehyde before coating with gold. Dehydration was done with an ethanol series at 4˚C

using the critical point drying. SEM imaging analysis was performed with JEOL InTouch-Scope™ series SEMs JSM-IT500HR using 10 kV for all magnifications.

## Swelling test

For water absorption capacities of SF/G corneal films and SF/G stromal scaffolds, swelling was measured. Dry forms of SF/G corneal films and SF/G stromal scaffolds were weighed and then immersed in DI water at for a specific amount of time. Before weighing again, the materials were blotted with paper to remove excess water. The water content was then calculated with the following equation:

$$\text{Water absorption} = (M_t - M_0)/M_t \, X \, 100\%$$

where $M_0$ is the initial weight of materials and $M_t$ is weight of wet form at a specific point in time.

## *In vitro* enzymatic degradation

*In vitro* degradation tests were used to evaluate the degradation rate of SF/G corneal films and SF/G stromal scaffolds at specific time points. Briefly, SF/G corneal films and SF/G stromal scaffolds were incubated in 1U/mL of protease XIV solution (pH 7.4) (Sigma-Aldrich Corporation, USA) containing 0.01% w/v sodium azide (Loba Chemie PVT. LTD., India) for antimicrobials at 37˚C and replaced every 2 days. Hereafter, the remaining materials were washed with DI and centrifuged at 10,000 rpm for 5 min, 3 times to remove non-material solution. Materials were dried at 60˚C in a heating block overnight. Then, the dried form of the materials was weighed and normalized according to following equation:

$$\text{Residual mass } (\%) = W_t/W_0 \, x \, 100\%$$

where $W_0$ is the initial weight of materials and $W_t$ is the weight of materials at specific point in time.

## Uniaxial tensile test

A Shimadzu universal testing machine was used to determine tensile strength under a cross-head speed of 5 mm/min until ultimate failure. 40-μm-thick SF/G corneal films and 1-mm-thick SF/G stromal scaffold were cut into dumbbell shapes with 2 cm of gag length and 1-cm width and immersed in PBS until equilibrium. The ultimate tensile strength (UTS), elastic modulus (Mpa), and elongation values at break were automatically calculated.

## Canine limbal epithelial stem cells (cLESCs) isolation and culture

This project used canine cadaveric corneas, so it was not applicable for IACUC approval. Unilateral/Bilateral corneas were obtained from 15 cadaveric healthy dog eyes (aged between 1 to 7 years old) without pathological lesion of the eyes and less than 4 h of preservation. Subject information is provided in S1 Table. Briefly, corneas were rinsed with Hank's balanced salt solution (HBSS) (Thermo Fisher Scientific Corporation) containing 100 unit/mL penicillin (Thermo Fisher Scientific Corporation), 100 μg/mL streptomycin (Thermo Fisher Scientific Corporation), 5 μg/mL amphotericin B (Thermo Fisher Scientific Corporation), and 50 μg/mL gentamycin (Thermo Fisher Scientific Corporation) 6 times. After removing the excess tissue, corneas were dissected at the superficial limbal area into 8 pieces equally using a crescent knife (Alcon, USA) and corneal scissors and incubated in collagenase I (Sigma-Aldrich Corporation, USA) at 0.5 mg/mL for 16 h at 37˚C. After that, epithelial cells were dissociated with a cell

scraper on the epithelial side and pipetting to create a single cell suspension. The isolated cells were seeded on the prepared plate with mitotically inactive 3T3 J2 fibroblast. Limbal epithelial stem cell media containing high-glucose Dulbecco's Modified Eagle Media: Nutrient Mixture F-12 (DMEM/F-12; Thermo Fisher Scientific Corporation) supplemented with 10% fetal bovine serum (FBS; Thermo Fisher Scientific Corporation), 2 mM L-glutamine (100x Gluta-MAX™; Thermo Fisher Scientific Corporation), 100 unit/mL penicillin (Thermo Fisher Scientific Corporation), 100 μg/mL streptomycin (Thermo Fisher Scientific Corporation), 5 μg/mL amphotericin B (Thermo Fisher Scientific Corporation), 20 ng/mL human recombinant epidermal growth factor (EGF; Millipore Corporation), 1x insulin-transferrin-selenium (ITS; Invitrogen), and 0.5 μg/mL hydrocortisone (Sigma-Aldrich Corporation, USA) was used for culture at 37°C under 95% humidity and 5% $CO_2$. Media were substituted every 2 days. Passage 3 cells were used for further experiments.

### Preparation of 3T3 feeder cells

3T3 J2 fibroblast cells were kindly given to the team by the Stem Cell and Cell Therapy Research Unit, King Chulalongkorn Memorial Hospital. After confluence, 3T3 proliferation was inhibited with 10 μg/mL of mitomycin C (Sigma-Aldrich corporation, USA) for 2 h at 37°C under 5% $CO_2$ and washing with PBS 2 times before seeding the cLESCs.

### Canine corneal stromal stem cells (cCSSCs) isolation and culture

Canine corneal stromal stem cells (CSSCs) were isolated after removing limbal epithelial stem cells. The remaining limbal tissues were cut into smaller pieces and scraped with a cell scraper (SPL Lift Sciences, Korea). Then, the tissues and cell suspension were filtered through a 70-μm cell strainer (SPL Lift Sciences) and centrifuged to obtain a cell pellet. The pellet was resuspended and seeded in a 60-mm culture plate. Corneal stromal stem cell proliferation media contained DMEM/MCDB-201 (1:1 (v/v); Sigma-Aldrich corporation, USA), 2% fetal bovine serum, 10 ng/mL platelet-derived growth factor (PDGF-BB; Millipore corporation), 10 ng/mL epithelial growth factor (EGF; Millipore corporation), 0.1 mM ascorbic acid, $10^{-8}$ M dexamethasone, and 1x insulin-transferrin-selenium (ITS; Invitrogen) and was cultured at 37°C under 5% $CO_2$. The third passage was used in further experiments.

### Keratocyte differentiation

cCSSCs at the third passage were seeded into SF/G stromal scaffold at a concentration of $6x10^6$ cells/mL; 80 μl for construction of the corneal stromal part and seeded into the culture plate for characterization. After 3 days, the media were changed from corneal stromal stem cell proliferation media to keratocyte differentiation media (KDM), consisting of DMEM (Thermo Fisher Scientific Corporation) supplemented with 1.0 mM L ascorbic acid-2-phosphate (Sigma-Aldrich Corporation, St Louis, USA), 2 mM L-glutamine (100x GlutaMAX™; Thermo Fisher Scientific Corporation), 100 unit/mL penicillin (Thermo Fisher Scientific Corporation), 100 μg/mL streptomycin (Thermo Fisher Scientific Corporation), 5 μg/mL amphotericin B (Thermo Fisher Scientific Corporation), 10 ng/mL basic fibroblast growth factor (FGF-2, Millipore corporation), and 0.1 ng/mL transforming growth factor-beta3 (TGF-ß3, Sigma-Aldrich Corporation, USA).

### Quantitative reverse transcription PCR (RT-qPCR)

RT-qPCR was performed to characterize both cLESCs and cCSSCs 14 days after culturing in tissue culture plates (TCP) for characterization. cLESCs seeded on SF/G corneal film and

cCSSCs seeded into SF/G stromal scaffold were executed at day 14 and day 28. Briefly, RNA was collected by using TRIzol-RNA isolation reagent (Thermo Fisher Scientific Corporation, USA) followed by extracting with the Direct-Zol RNA isolation kit (Zymo Research, USA) according to the manufacture's protocols. The ImProm-$^{TM}$ Reverse Transcription System (Promega, USA) was used for converting total RNA to cDNA. The targeted genes were amplified by FastStart Essential DNA Green Master (Roche Diagnostics) using CFX96™ real-time PCR detection system (Bio-Rad). In this experiment, glyceraldehyde 3-phosphate dehydrogenase (*Gapdh*) was appointed to be a reference gene. The gene expression levels were normalized by *Gapdh* and calculated by this formula: $2^{-\Delta\Delta CT}$, where $\Delta\Delta CT = [Ct^{target\ gene}$-$Ct^{GAPDH}]_{treated}-[Ct^{target\ gene}-Ct^{GAPDH}]_{control}$ (Ct referred to cycle threshold).

## Cell proliferation assay

For cell proliferation, the DNA of cLESCs seeded in SF/G corneal film (100,000 cell/cm$^2$) and cCSSCs seeded in SF/G stromal scaffold (cell concentration 6x10$^6$ cells/mL; 80 μl/scaffold) were quantified at days 1, 3, 5, 7, 14, and 28. The samples of each time point were digested with 1.0 mg/mL proteinase K (Worthington Biochemical, USA) in proteinase K buffer (150 mM Tris HCl and 1 mM EDTA pH 8.0) and 20 μl/sample of papain suspension (Worthington Biochemical, USA). After that, all samples were incubated at 60˚C for 16 h in a heating block. Centrifugation was performed with 8,000 rpm for 10 min at 4˚C to get the supernatant and transferred to a new tube. DNA levels were measured by Qubit Fluorometric Quantification with dsDNA BR assay according to the manufacture's protocols. Quantitative data was normalized with plain film and scaffold.

## Cell viability assay and distribution

Parallel to the time point of the cell proliferation assay, LIVE/DEAD cell double staining was illustrated by cLESCs seeded SF/G corneal films and cCSSCs seeded SF/G stromal scaffolds. 100,000 cell/cm$^2$ of cLESCs seeded in SF/G corneal film and 6x10$^6$ cells/mL; 80 μl/scaffold of cCSSCs seeded in SF/G stromal scaffold were used. The samples were washed with HBSS (Thermo Fisher Scientific Corporation, USA) to remove nonspecific background. Before staining, calcein AM solution (Thermo Fisher Scientific Corporation, USA) and propidium iodide solution (Sigma-Aldrich Corporation, USA) were prepared by diluting with HBSS (1:1,000 in HBSS; Thermo Fisher Scientific Corporation, USA) simultaneously. The staining solution was then added and incubated for 30 min at 37˚C. After washing with HBSS (Thermo Fisher Scientific Corporation, USA), the samples were observed under a fluorescent microscope incorporated with Carl Zeiss$^{TM}$ ApoTome.2 apparatus (Carl Zeiss, Germany).

## Bio-fabricated cornea formation

To mimic 3-D corneal structure, both materials were placed on a 12-well cell insert (translucent PET high pore density (HD); Falcon corporation) separately. cLESCs were cultured with limbal epithelial stem cell media on SF/G corneal film, whereas cCSSCs in SF/G stromal scaffolds were cultured with corneal stromal stem cell proliferation media for 3 days. For assembly, tissue glue (Tisseel®; Baxter corporation, USA) was applied on cCSSCs seeded SF/G stromal scaffolds. cLESCs seeded SF/G corneal films were then gently transferred on the top and left in RT for 5 min to complete polymerization. Then, media were added with keratocyte differentiation media fully covering the whole scaffolds and further cultivated for 7 days. Thereafter, an air lifting interface was generated to induce epithelial cell stratification. Media were resubstituted every 2 days until 14 and 28 days.

## Histology, immunocytochemistry and immunohistochemistry

Routine H&E straining was used to evaluate cell morphology and distribution onto/into scaffolds. For immunocytochemistry preparation, cLESCs and cCSSCs were seeded into a chamber slide (SPL Life Sciences, Korea) with 5,000 cells/chamber and incubated at 37˚C under 95% humidity and 5% $CO_2$ for 24 h. Then, the slides were fixed with 100% cold methanol for 15 min and permeabilized with 0.1% Triton X-100 (Sigma-Aldrich, Missouri, USA) for 10 min. To reduce non-specific background, 10% donkey serum in PBS was used for 1 h. The primary antibodies were incubated for characterized cLESCs and cCSSCs using mouse monoclonal anti-P63 (1:50; ab735, Abcam) and rabbit polyclonal anti-Pax 6 (1:500, 901301, BioLegend), respectively, at 4˚C overnight.

For immunofluorescence, cLESCs seeded SF/G corneal films at day 14 and 28 were transferred to the chamber slide (SPL Life Sciences) and prepared as the aforementioned procedure. cCSSCs seeded SF/G stromal scaffolds and bio-fabricated cornea at day 14 and 28 were washed with PBS and fixed in 4% paraformaldehyde. After dehydration with alcohol, samples were embedded in paraffin blocks and cut into 3-µm-thick sections using a microtome and transferred to slides. After that, the sections were rehydrated in xylene and through descending graded series of alcohol. 5-min microwave heating (600 w) was performed twice in citric acid buffer (pH 6) for antigen retrieval. Then, the sections were blocked in 10% donkey serum in PBS for 1 h and incubated with primary antibodies diluted with PBS and 1% bovine serum albumin (BSA; Sigma-Aldrich Corporation, USA) overnight at 4˚C. For primary antibody, to characterize and localize cLESCs and differentiated keratocytes, mouse monoclonal anti-P63 (1:50; ab735, Abcam) and rabbit polyclonal anti-Aldh3a1 (1:200; ab129815, Abcam) were applied, respectively. For extracellular matrix investigation, rabbit monoclonal anti-lumican (1:200; ab168348, Abcam) and rabbit polyclonal anti-Collagen I (1:400; ab254113, Abcam) were used to determine lumican and collagen production. All antibodies were incubated separated in the adjacent sections.

For secondary antibodies, FITC-conjugated goat anti-mouse IgG H&L (1:1,000; ab6785, Abcam) and Cy3-conjugated goat anti-rabbit (1:500; minimal x-reactivity, BioLegend) were then used at RT in the dark room for 1h. DAPI was used for nuclear counterstain. The sections were then mounted with Vectashield (Vector Laboratories Inc., Burlingame, CA). Isotype controls served as the negative control. For visualization, a fluorescent microscope incorporated with Carl Zeiss™ Apotome.2 apparatus (Carl Zeiss, Germany) was used.

## Statistical analysis

Data analysis was represented as dot plot (N = 4) using GraphPad Prism 9.0 (GraphPad Software, San Diego, CA), and statistical analysis was determined using SPSS statistics 22 software (IBM Corporation, USA). ImageJ was used to analyze and quantify the images. For comparing the 2 groups and more than 2 groups of continuous parameters, nonparametric tests were used with the Mann-Whitney test and Kruskal-Wallis test, respectively. The significance level was considered to be when $p$ was <0.05. All experiments were run in quadruplicate for independent experiments.

## Results

### cLESCs and cCSSCs isolation and characterization

cLESCs and cCSSCs were successfully isolated from the canine cadaveric cornea with collagenase I. cLESCs were formed holoclones within 3 days after isolation. Cobble-stone-like cell sheets surrounded with the area of interface and 3T3 inactive 3T3 J2 fibroblast were presented

(Fig 1A). cLESCs exhibited stemness-related markers (*Rex 1 and Oct4*) (Fig 1C), proliferation marker (*Ki67*) (Fig 1D), limbal stem cell markers (*Abcg2* and *P63*) (Fig 1E), and corneal epithelial markers (*Krt3 and Krt12*) (Fig 1F). cCSSCs were also adhered to culture plate 2 h after seeding and reached 60–70% confluence within 7 days. The morphology turned from an oval shape to a fibroblast-like appearance (Fig 1B). Gene expression of cCSSCs indicated both pluripotent and adult stemness-related markers (*Rex 1*, *Oct4*, *Pax6*, *Ngfr*, *Nes* and *CD90*) (Fig 1G and 1J), proliferation marker (*Ki67*) (Fig 1H), genes associated with keratocytes (*Aldh3a1*, *Aqp1*, *Kera*, and *Lum*) (Fig 1K), and myofibroblast marker (*ACTA2*) (Fig 1I). Moreover, tumor protein P63 and Pax6 (oculorhombin), which are the markers of cLESCs and cCSSCs respectively, were presented by immunocytochemistry staining (Fig 1L and 1M) [27, 28].

In addition, cCSSCs were characterized by differentiation into keratocytes using KDM in culture plate. After differentiation, cCSSCs changed their morphology from a small oval and spindle shape to a stellate and elongated shape (Fig 2A and 2B). Gene expression of cCSSCs and differentiated keratocytes revealed significant downregulation of proliferation (*Ki67*)

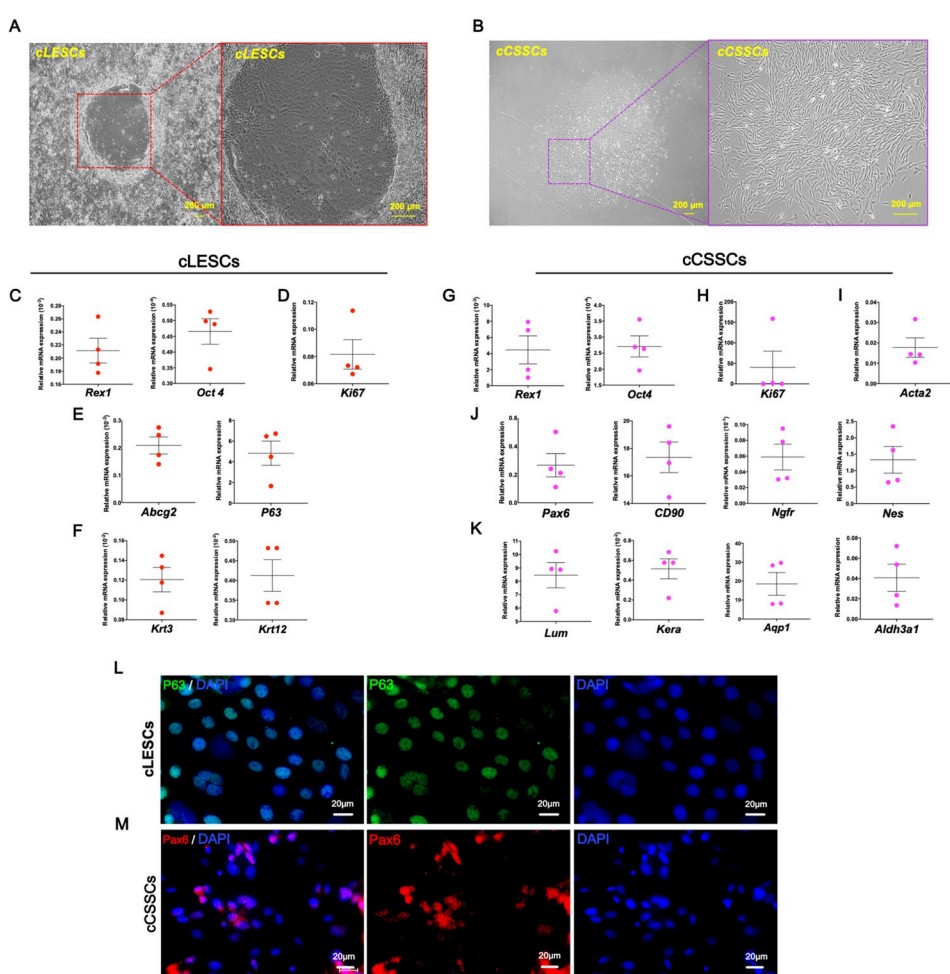

**Fig 1. cLESCs and cCSSCs characterization.** The morphological appearances of cLESCs (A) and cCSSCs (B) are illustrated under a phase contrast microscope with 40x and 100x magnification. cLESCs and cCSSCs mRNA expression are quantified by RT-qPCR and normalized by *Gapdh*. Pluripotent stemness-related markers (C, G), adult stemness-related markers (E, J), proliferation marker (D, H), myofibroblast marker (I), epithelial cell markers (F), keratocyte marker (K) are characterized (n = 4). Immunocytochemistry of cLESCs and cCSSCs expresses P63 and Pax6 respectively (L, M) with 400x magnification. The scale bars present 200 μm (A, B) and 20 (L, M).

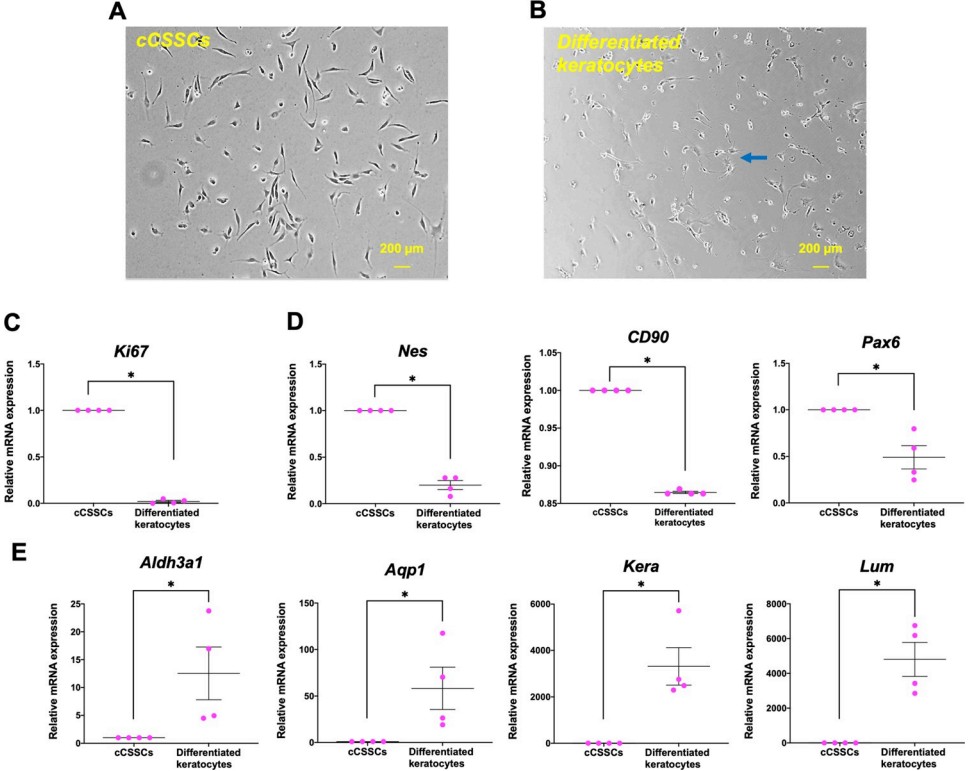

**Fig 2. cCSSCs and differentiated keratocytes.** Phase contrast imaging of cCSSCs presents a small oval and spindle shape with 40x magnification (A). Cell morphology illustrates an elongated and stellate shape of differentiated keratocytes with 40x magnification, pointed out by the blue arrow (B). The scale bars present 200 μm (A, B). cCSSC and differentiated keratocyte mRNA expressions are quantified by RT-qPCR and normalized by *Gapdh*. Proliferation marker (C), adult stemness-related markers (D), keratocyte markers (E) are characterized (n = 4) (*$p < 0.05$).

(Fig 2C) and adult stemness-related markers (*Nes*, *CD90*, *Pax6*) (Fig 2D), whereas there was a significant upregulation of genes associated with keratocytes (*Aldh3a1*, *Aqp1*, *Kera*, and *Lum*) (Fig 2E) ($p<0.05$).

The results suggested that cLESCs and cCSSCs were isolated, cultured, and expanded *in vitro*, according to our established protocol.

## Bio-fabrication and characterization of silk fibroin/gelatin (SF/G)-based corneal film and stromal scaffold

In the matter of SF/G corneal film, the substrate of cLESCs was constructed by casting from 2-mL SF/G solution, resulting in an average thickness of 40 μm. The dry and wet forms' transparency exhibited a weave texture pattern and color of the background behind (Fig 3A and 3B). In comparison, the wet form provided superior clarity due to water absorption (Fig 3B). From SEM evaluation, the surface of SF/G corneal film was absolutely smooth and non-porous (Fig 3C). In terms of the SF/G stromal scaffold, the appearance revealed a coin-like, opaque white shape (Fig 3D). However, after water equilibrium, transfiguration to a white faded color was observed (Fig 3E). The process of freeze drying achieved a generalized production of small pores in the material. The surface and cross-sections illustrated multiple pores and irregular surface topography. The pores were sharp edged with an approximate diameter of 130.71 ± 37.12 μm (Fig 3F).

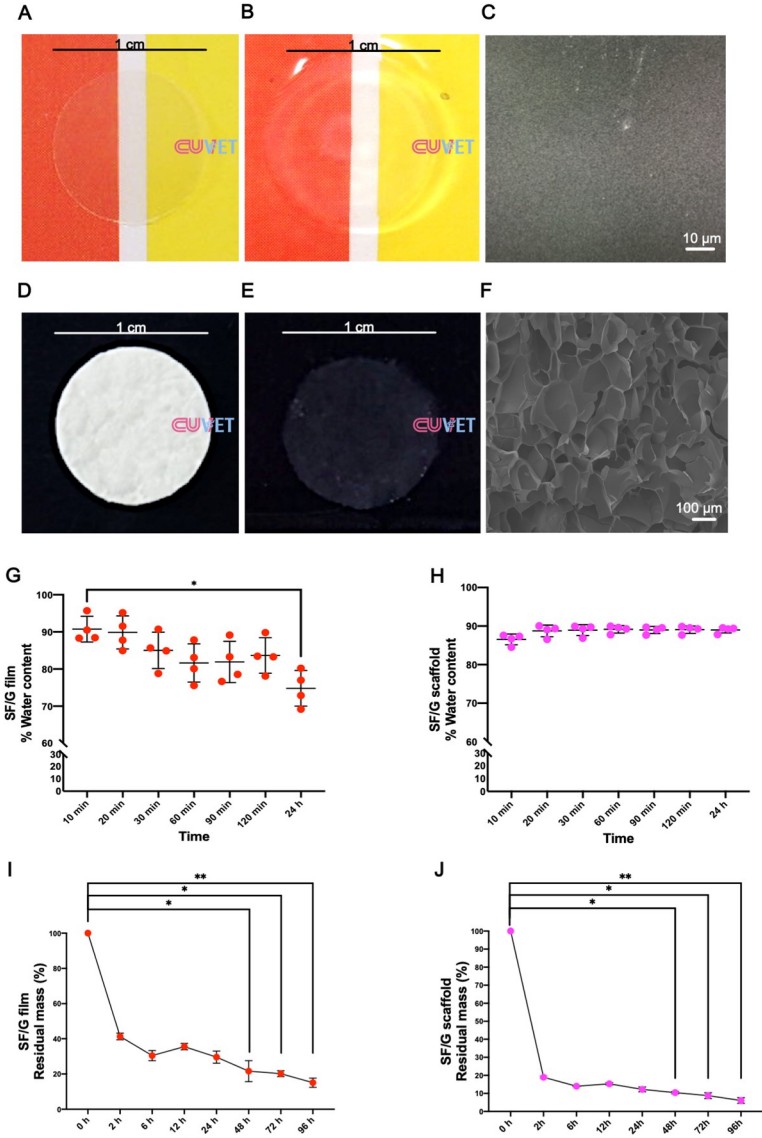

**Fig 3. SF/G corneal film and scaffold characterization.** The external appearance of SF/G corneal film in dry form (A) and wet form (B) as well as SF/G stromal scaffold in dry form (D) and wet form (E) are illustrated. With regard to ultrastructure features, scanning electron microscope shows a smooth surface for the SF/G corneal film (1000x magnification) (C). A 100x magnification view of SF/G stromal scaffold shows a porous surface (F). Equilibrium of water content related to swelling capacity of SF/G corneal film (G) and SF/G stromal scaffold (H) are presented. Enzymatic *in vitro* degradation rate of SF/G corneal film (I) and SF/G stromal scaffold (J) using protease XIV solution are also presented. Data are from 4 independent experiments (n = 4) ± standard deviation ($^{*}p<0.05$, $^{**}p<0.01$). The scale bars present 1 cm (A, B, D, E), 10 μm (C) and 100 μm (F).

The results illustrated the morphological characteristics of the bio-fabricated material that are suitable for use as corneal film and stromal scaffold.

## Physical properties of SF/G corneal film and stromal scaffold

The swelling tests of SF/G corneal films exhibited more than 90% water content and gradually decreased at any time point until a significant difference at 24 h ($p<0.05$) (Fig 3G). Meanwhile, SF/G stromal scaffold had a capacity to sustain its % of water content further than 85% and no

significant difference at all time points (Fig 3H). *In vitro* enzymatic degradation manifested less than 50% of residual mass in both materials within 2 h. For SF/G corneal film, the % of residual mass was between 20–40% during 6 h to 72 h and less than 20% at 96 h (Fig 3I). Additionally, for the SF/G stromal scaffold, the % of residual mass remained 10–20% after 2 h up to 48 h, after which it was almost entirely degraded with less than 10% (Fig 3J). Significant differences were shown between initial time point and 48 h ($p<0.05$), 72 h ($p<0.05$), and 96 h ($p<0.01$) of both materials (Fig 3I and 3J). Mechanical testing determined tensile characteristics of both materials, and the results revealed remarkable differences described by elastic modulus, ultimate tensile strength value (UTS), and % elongation at break. The elastic modulus of SF/G stromal scaffolds, UTS, and % elongation at break was lower than the other. However, significant differences were not noticed (Table 1).

The evidence suggested the physical properties favored the application as cell-incorporated biomaterial.

**SF/G corneal film and stromal scaffold favor cell biocompatibility.** According to the morphological and physical characteristics of the SF/G corneal film and stromal scaffold, specific stem cell populations, cLESCs and cCSSCs were respectively seeded onto the materials. The biocompatibilities of stem-cell-incorporated SF/G corneal film and stromal scaffold were analyzed.

SEM imaged the morphology of cLESCs seeded SF/G corneal film and cCSSCs seeded SF/G stromal scaffold from a top view. cLESCs were feasibly adhered and aligned all over the SF/G corneal film area. The surface of the cLESCs revealed a polygonal cell shape and proximity to each other. 2–3 layers of stratified squamous epithelium were presented. The superficial cells exhibited bigger and less connection space of cell wall than basal cells (Fig 4A). SF/G stromal scaffold provided a comprehensive habitat for cCSSCs. SEM detected those cells had grown onto and into the pores of the sponge-like scaffold. The cells were spread out and distributed throughout the scaffold area. A fibrin-like extracellular matrix produced by cCSSCs was observed on the surface and in the pores (Fig 4B).

Long term cell viability and distribution of cLESCs seeded SF/G corneal film and cCSSCs seeded SF/G stromal scaffold were determined at days 1, 3, 5, 7, 14, and 28 and described by intracellular esterase activity and cell membrane permeability. A large number of cells at all time points survived, but a few dead cells were indicated by green color at the cytoplasm and red nucleus, respectively (Fig 4C and 4D). cLESCs and cCSSCs displayed plentiful distribution on SF/G corneal films and SF/G stromal scaffold at all time points, respectively. In addition, cCSSCs seeded SF/G stromal scaffold demonstrated a majority of its cells' viability and adhesion inside and on the bottom of scaffold (Fig 4E and 4F).

DNA measurement defined as cell proliferation assay was investigated using Qubit Flex Fluorometer. The dot plot graphs of DNA quantitation of cLESCs seeded SF/G corneal film and cCSSCs seeded SF/G stromal scaffold were indicated cell proliferation at days 1, 3, 5, 7, 14, and 28. cLESCs seeded in SF/G corneal film increased proliferation at days 1, 3, and 5 and were rather unchanged after day 5 to day 28. Summarized data determined a significant difference between day 1 and day 28 ($p<0.05$) (Fig 4G). In cCSSCs seeded SF/G stromal scaffold, a stable proliferation rate was manifested at days 1, 3, 5, and 7 and obviously increased after day

**Table 1. Elastic modulus, UTS, and strain at UTS.**

| Materials | Elastic modulus (Mpa) | UTS (Mpa) | Elongation (%) |
|---|---|---|---|
| SF/G corneal film | 1.77±0.40 | 0.56±0.18 | 68.4±2.98 |
| SF/G stromal scaffold | 1.25±0.91 | 0.18±0.09 | 2.63±0.58 |

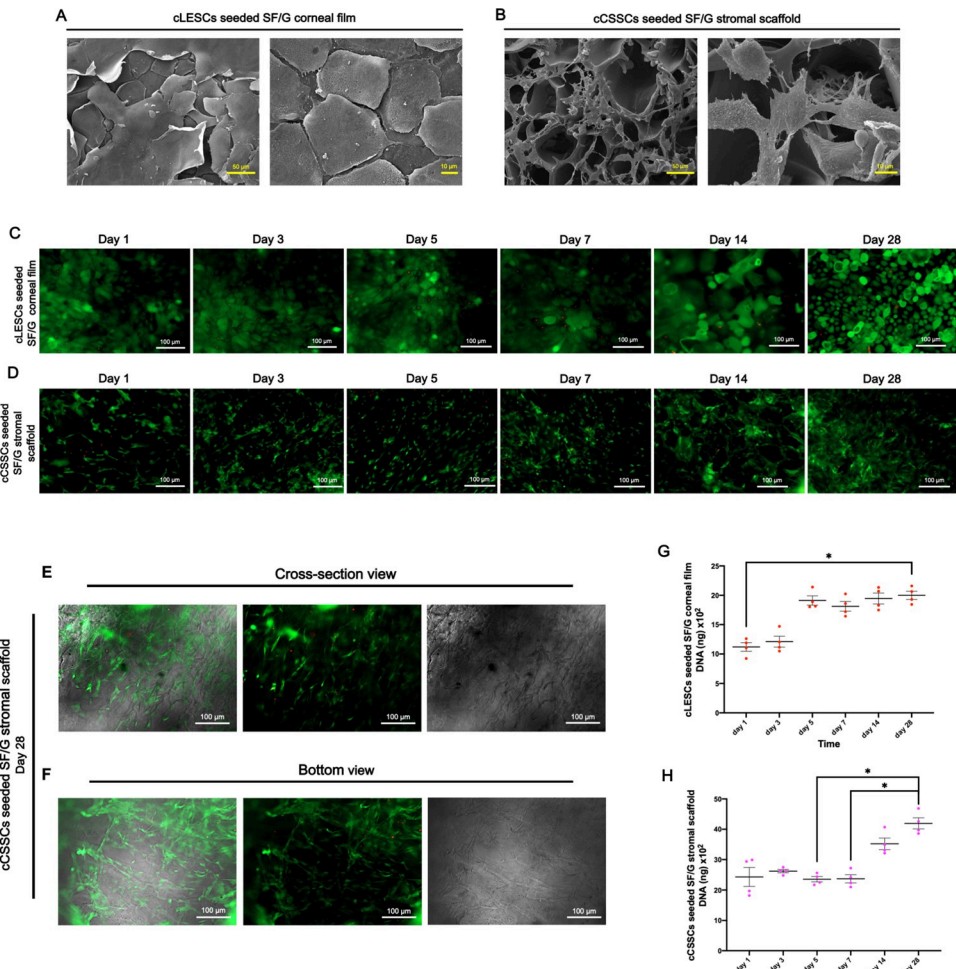

**Fig 4. cLESCs seeded SF/G corneal film and cCSSCs seeded SF/G stromal scaffold.** Scanning electron microscope (SEM) images of cLESCs seeded SF/G corneal film (350x and 1000x magnification) (A) and cCSSCs seeded SF/G stromal scaffold (350x and 1500x magnification) (B) at 14 days of culturing. Fluorescent images of live/dead staining of cLESCs seeded SF/G corneal film (C) and cCSSCs seeded SF/G stromal scaffold (D) at days 1, 3, 5, 7, 14, and 28 (100x magnification). Fluorescent images of live/dead staining, brightfield (BF), and merges at the cross-section view (E) and the bottom view (F) of cCSSCs seeded SF/G stromal scaffold at day 28 (100x magnification). Cell proliferation assay shows the proliferation pattern of cLESCs seeded SF/G corneal film (G) and cCSSCs seeded SF/G stromal scaffold (H) at days 1, 3, 5, 7, 14, and 28. All the data are expressed as means ± standard error of mean (SEM) (n = 4) ($^*p < 0.05$). The scale bars present 50 and 10 μm (A, B) and 100 μm (C, D, E, F).

7 to day 28. DNA levels were significantly different between day 5 and 28 ($p < 0.05$) and day 7 and 28 ($p < 0.05$) (Fig 4H).

The results illustrated that SF/G corneal film and stromal scaffold provided favorable biocompatibility for cLESCs and cCSSCs.

**SF/G corneal film and stromal scaffold promote maturation mRNA marker expressed by cLESCs and cCSSCs.** To evaluate the beneficial properties of SF/G corneal film and stromal scaffold for cell properties, mRNA marker expression by cLESCs and cCSSCs was analyzed using RT-qPCR. The results were presented by dot plot graph among day 14 of cells seeded on tissue culture plates (TCP; control) and day 14 and day 28 of cells seeded on their own materials. Data of day 14 and day 28 were normalized by the control. For cLESCs seeded SF/G corneal film, *Rex1* and *Oct4* revealed higher gene expression than the control. *Rex1* at

day 14 was significantly increased ($p<0.05$), but no significant increase was presented at day 28 compared to the control (Fig 5A). *Oct4* was significantly increased from the control to day 28 ($p<0.05$) (Fig 5A). *Ki67* at day 14 was notably higher than the control but significantly lower from day 14 to day 28 ($p<0.01$) (Fig 5B). For *Abcg2* and *P63*, levels of gene expression at day 14 and day 28 were higher than the control, respectively. Significant upregulations appeared between the control and day 14 of *Abcg2* ($p<0.01$) (Fig 5C). *Krt3* and *Krt12* were

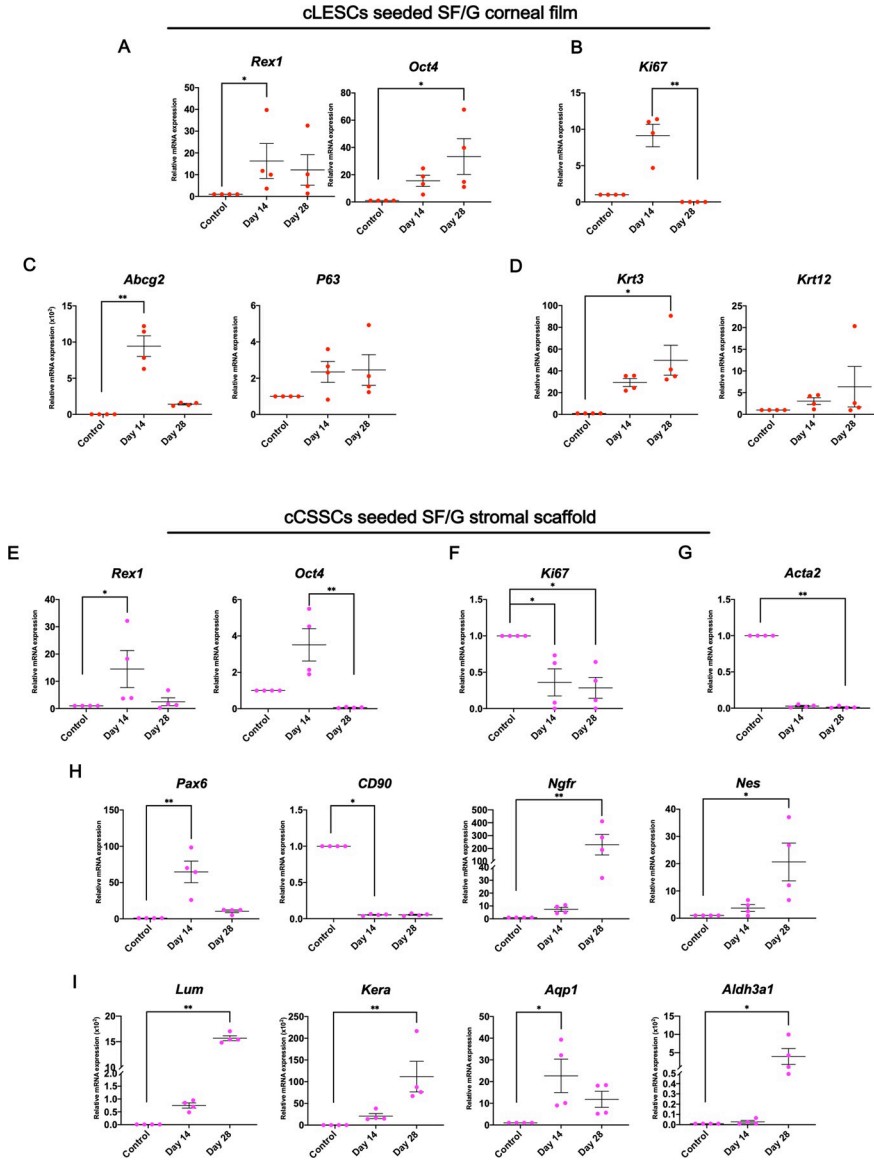

**Fig 5. mRNA expression of cLESCs seeded SF/G corneal film and cCSSCs seeded SF/G stromal scaffold.** Fold changes of RT-qPCR expression data of the control (cells culture on tissue culture plate for 14 days) and cLESCs seeded SF/G corneal film at 14 and 28 days are presented. cLESCs seeded SF/G corneal film at 14 and 28 days shows the upregulation of pluripotent (A), adult stemness-related markers (C), and epithelial cells markers (D) compared to the control. The proliferation marker shows upregulation at day 14 and downregulation at day 28 (B). RNA expression of pluripotent (E) and adult stemness-related markers (H) of cCSSCs seeded SF/G stromal scaffold reveals upregulation at day 14, except CD90 shows downregulation at day 14 and 28. The proliferation marker (F) and myofibroblast marker (G) are downregulated at both time points. Upregulation at day 14 and 28 of both time points is presented by the keratocyte marker (I). Values represent mean ± SEM, n = 4 ($^{*}p < 0.05$, $^{**}$p<0.01).

increased compared to the control, with significant upregulation of *Krt3* at day 28 ($p<0.05$) (Fig 5D).

cCSSCs seeded SF/G stromal scaffold displayed upregulation of *Rex1* and *Oct4* at day 14 but downregulation at day 28. A significant difference was detected between the control and day 14 ($p<0.05$) and between day 14 and day 28 ($p<0.01$) of *Rex1* and *Oct4*, respectively (Fig 5E). *Ki67* at day 14 and day 28 demonstrated significant downregulation compared to the control ($p<0.05$) (Fig 5F). *Pax6* expression showed a similar pattern to stemness-related markers with a significant increase at day 14 ($p<0.01$) (Fig 5H). Downregulation of *CD90* was illustrated both day 14 and day 28 ($p<0.05$) (Fig 5H). *Nes* and *Ngfr* were upregulated by time and significantly different at day 28 (Fig 5H). *Acta2* was significantly downregulated at both time points and significantly different at day 28 ($p<0.01$) (Fig 5G). The *Aldh3a1*, *Aqp1*, *Kera*, and *Lum* of day 14 and day 28 were higher than the control, particularly *Kera* and *Lum*, with a more than 1,000-fold elevation presented at day 28 ($p<0.01$) (Fig 5I). *Aqp1* level was significantly higher at day 14 ($p<0.05$) but insignificantly decreased from day 14 to day 28. Significant upregulation was illustrated at day 28 for *Aldh3A1* ($p<0.05$) (Fig 5I).

SF/G corneal films and stromal scaffolds proved to be acceptable substrates of cLESCs and cCSSCs in terms of promoting limbal stemness and keratocyte markers.

## Assembly of stem-cell-incorporated corneal epithelial and stromal equivalents derived from SF/G corneal film and stromal scaffold

To generate stem-cell-incorporated corneal epithelial and stromal equivalents aimed for canine corneal regeneration, cLESCs seeded SF/G corneal film and cCSSCs seeded SF/G stromal scaffold were assembled to imitate natural cornea. cCSSCs seeded SF/G stromal scaffold, acting as a corneal stroma, was covered by a synthetic corneal epithelium using cLESCs seeded SF/G corneal film with tissue glue (Tisseel®; Baxter Corporation, USA). The procedure is presented by the infographic diagram (Fig 6A).

After assembling, bio-fabricated cornea revealed a 1-cm-diameter coin-like structure. Its opacity was slightly increased compared to cLESCs seeded SF/G corneal film and cCSSCs seeded SF/G stromal scaffold; nevertheless, the background paper color was visible (Figs 3B, 3E, and 6B). cLESCs seeded SF/G corneal film was completely overlaid on cCSSCs seeded SF/G stromal scaffold without detachment (Fig 6B).

To examine the bio-fabricated cornea, SEM was used to illustrate the integrated bio-materials at day 14 after co-culture. Cross-sectional figure demonstrated the upper part of cLESCs seeded SF/G corneal film adhered to the lower part of cCSSCs seeded SF/G stromal scaffold. Additionally, the interconnected space was 12.71 ± 6.3 µm (Fig 6D). SF/G corneal films maintained a stable structure and smooth substrate for cLESCs. Flat squamous cLESCs were contentedly aligned on the SF/G corneal film in approximately 2–3 layers (Fig 6C and 6D). In addition, SF/G stromal scaffolds were able to sustain their construction with no evidence of collapsed pores. cCSSCs were infiltrated and dispersed into the sponge-like scaffold pores. Cell attachment was observed within each pore as well as cross-linkage to nearby pores (Fig 6C). Interestingly, the interconnected space was incorporated by tissue glue residue and ECM production (Fig 6D).

Our established technique successfully generated stem-cell-incorporated corneal epithelial and stromal equivalents for canines.

**Sequential characterization of stem-cell-incorporated corneal epithelial and stromal equivalents for canine corneal regeneration.** To explore the morphological, histological, and biocompatible properties of the generated stem-cell-incorporated corneal epithelial and stromal equivalents, sequential characterization using histological and immunohistochemistry

A

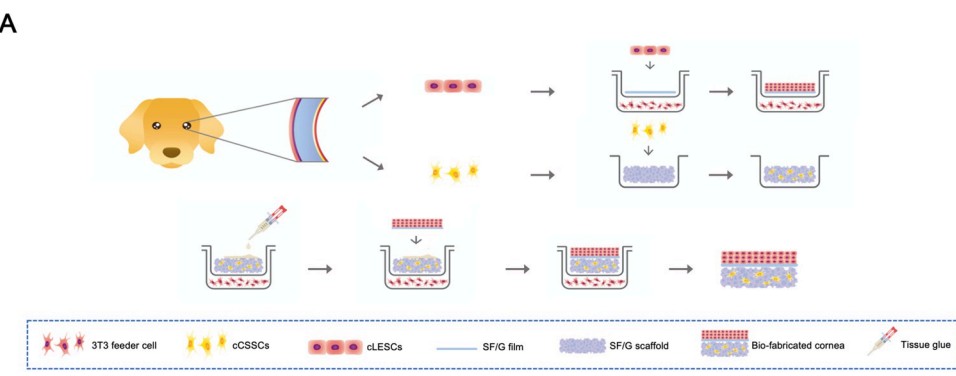

## Bio-fabricated cornea

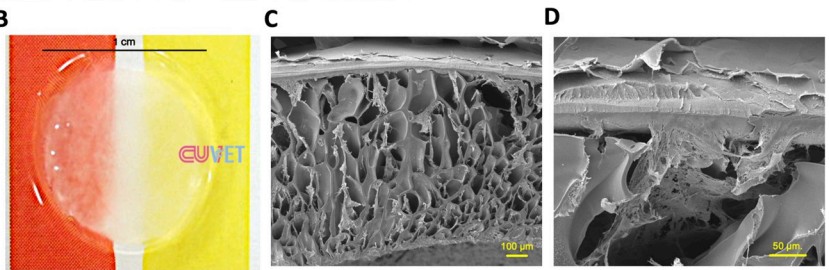

**Fig 6. Bio-fabricated cornea.** Schematic illustration of the process to generate bio-fabricated cornea by assembling of cLESCs seeded SF/G corneal film and cCSSCs seeded SF/G stromal scaffold (A). Gross appearance of bio-fabricated cornea (B). SEM shows the upper layer of cLESCs seeded SF/G corneal film aligned on the cCSSCs seeded SF/G stromal scaffold (100x (C) and 350x (D) magnification). The scale bars present 100 μm (C) and 50 μm (D).

techniques were employed. Analyses were performed at days 14 and 28 for cLESCs seeded SF/G corneal film, cCSSCs seeded SF/G stromal scaffold, and bio-fabricated cornea to evaluate P63 (cLESCs marker), lumican (ECM of keratocyte), and Aldh3a1 (marker of keratocyte). H&E investigated 2–3 layers of stratified cLESCs cultivated on days 14 and 28 of SF/G corneal films and bio-fabricated cornea. From the top view of cLESCs seeded SF/G corneal films, a large number of P63-representative cells were abundant and strongly presented at day 14 (Fig 7A), but these deceased at day 28 (Fig 7G). Moreover, cross-sectional bio-fabricated cornea detected P63-presenting cells, particularly at the basal layer at both time points (Figs 7D and 7J). cCSSC distribution and adherence along the surface of sponge-like scaffold pores was also investigated by H&E (Fig 7B and 7H). Additionally, bio-fabricated cornea manifested the two types of bio-material integration mimicry of natural cornea defined as the upper part of corneal epithelium and the lower part of corneal stroma (Fig 7D and 7J). However, the space between the 2 layers was revealed to be 108.94 ± 32.8 μm. At days 14 and 28, cCSSCs seeded SF/G stromal scaffold and bio-fabricated cornea showed fluorescent imaging of lumican and Aldh3a1 deposited by differentiated keratocytes (Fig 7B, 7C, 7E, 7F, 7H, 7I, 7K, and 7L). Comparing between 2 time points, the expression of lumican at day 28 was more robust and plentiful than day 14; on the other hand, Aldh3a1 expressions were alike (Fig 7B, 7C, 7E, 7F, 7H, 7I, 7K, and 7L).

The results illustrated that SF/G corneal films, SF/G stromal scaffolds, and bio-fabricated corneas were capable of imitating bona fide cornea by producing specific markers and exogenous ECM.

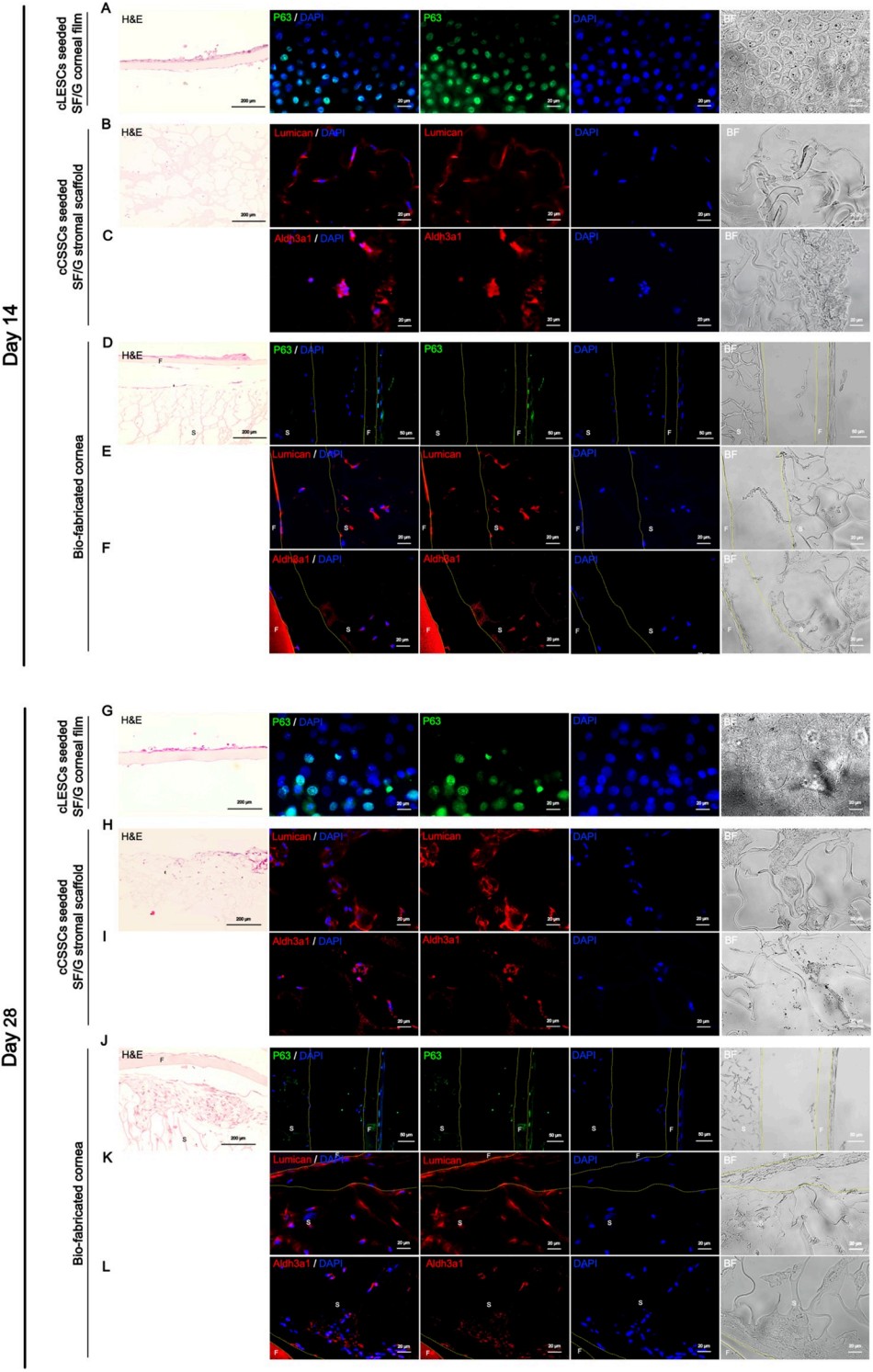

**Fig 7. Morphology and immunocytochemical profiles.** H&E staining shows 2–3 layers of cLESCs upon SF/G corneal film at day 14 (A) and day 28 (G) and the upper part of the bio-fabricated cornea at day 14 (D) and day 28 (J). cCSSCs were distributed over the SF/G stromal scaffold at day 14 (B) and day 28 (H) and the lower part of bio-fabricated cornea at day 14 (D) and day 28 (J). cLESCs seeded SF/G corneal film at day 14 (A) shows more P63-positive nuclear staining cells than day 28 (G). P63-positive nuclear staining cells are detected at the basal cells of the upper part of bio-fabricated cornea at day 14 (D) and day 28 (J). Lumican-positive cytoplasmic staining shows stronger intensity at day 28 for cCSSCs seeded SF/G stromal scaffold (H) and the lower part of bio-fabricated cornea (K) than day 14 of cCSSCs

seeded SF/G stromal scaffold (B) and the lower part of bio-fabricated cornea (E). Cytoplasmic staining of Aldh3a1 is similarly detected at days 14 and 28 of cCSSCs seeded SF/G stromal scaffold (C, I) and bio-fabricated cornea (F, L) respectively. H&E figures image with 200x magnification. Immunofluorescences image with 400x magnification of all figures, except the figures of P63 of bio-fabricated cornea image with 200x. F, S indicate the areas of SF/G corneal film and SF/G stromal scaffold respectively, which are separated by the yellow border. Scale bars present 200 μm of H&E staining images, 20 μm of immunofluorescences images (A, B, C, E, F, G, H, I, K, L), and 50 μm of immunofluorescences images (D, J).

**Stem-cell-incorporated corneal stromal equivalents promote collagen matrix deposition.** Collagen type I (Col-1), which is abundant in corneal stroma, was analyzed by immunohistochemical examination. cCSSCs seeded SF/G stromal scaffold revealed collagen type I fibrils deposited by differentiated keratocytes (Fig 8A and 8B). The quantitative collagen amount was determined using ImageJ and normalized by the area of scaffold. The result revealed a significant difference ($p < 0.05$) between 86.62 ± 2.38% of day 14 and 165.03 ± 9.79% of day 28. The data showed the elevation of collagen production according to the timeline. However, at day 14, collagen was being produced and could be used. The adhering cells, which were at the surface of the pore, generated collagen fibrils aligned along the scaffold (Fig 8A). In contrast, the cells that stayed inside the pore had less scaffold adherence and thus showed less potential to align directionally (Fig 8B).

In summary, collagen secretion was favorably presented in these corneal stromal equivalents, and collagen aligning was regulated by topographical modification of the scaffold.

## Discussion

Canine cornea consists of 3 cellular layers (corneal epithelium, corneal stroma, and corneal endothelium) and one interface called Descemet's membrane, which is a dense, thick, basement membrane for the corneal endothelium [6, 29–31]. Canine corneal ulcer is one of the most common corneal diseases causing pain and visual loss, resulting in low quality of life [5, 32]. Superficial corneal ulcer, deep corneal ulcer, spontaneous chronic corneal ulceration (SCCED), and descemetocele are common types of corneal ulcer that affect the part of the

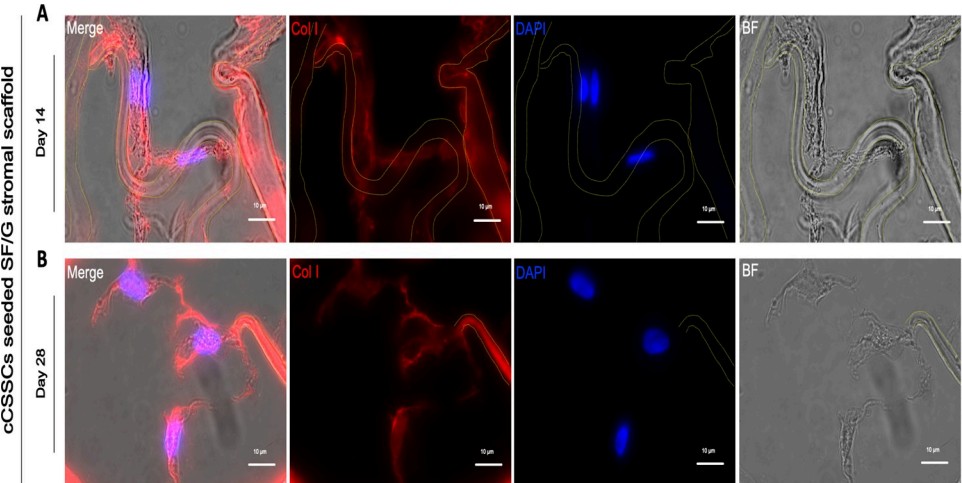

**Fig 8. Immunofluorescent staining of collagen I.** Differentiated keratocytes express collagen I at extracellular compartment at day 14 (A) and day 28 (B). Immunofluorescences image with 1000x magnification. Scaffolds are marginated by yellow border. Scale bars present 10 μm (A, B).

corneal epithelium and stroma [6]. Thus, canine corneal epithelial and stromal equivalents were generated as a transplantable graft using cell-incorporated SF/G scaffold.

A stem cell niche for corneal limbal epithelial stem cells (LESCs) is located in the limbal area called the corneoscleral junction. Corneal epithelial stem cells play an important role in self-renewal and corneal epithelial wound healing [33]. Enzymatic isolation of LESCs has typically been accomplished with dispase and trypsin [34]. However, now collagenase isolation provides superior efficiency because of its capacity to remove more basal epithelial progenitor cells and close-proximity mesenchymal cells and maintain some of the basement membrane matrix [35]. This study first determined the potential for collagenase isolation in canine LESCs described by holoclones formation within 3 days and expression of stemness-related markers (*Rex1* and *Oct4*), proliferation marker (*Ki67*), limbal stem cell markers (*Abcg2* and *P63*), and corneal epithelial markers (*Krt3* and *Krt12*). Moreover, P63α, widely recognized as a limbal stem cell marker in humans and canines, was illustrated in the majority of positive cell staining from the immunocytochemistry results [27, 36]. cCSCs (canine corneal stromal cells) are also a multipotent mesenchymal stem cell which possess stem cell marker expression (CD90, CD73, CD105, N-cadherin, and Pax6), trilineage differentiation, and innate immune privilege [28]. The expressions of *Rex 1*, *Oct4*, *Pax6*, *Ngfr*, *Nes* and *CD90* in this study fulfilled stem-cell-based knowledge of cell-derived canine corneal stromal stem cells. Transcription factor Pax6 (oculorhombin) was discovered in many embryonic ocular tissues, except keratocytes, and detected to be the marker of limbal stromal stem cells with nuclear immunostaining [37]. Unlike canine cells, human limbal niche cells manifested Pax6-positive nuclear staining but had negative exhibition in the nuclei of central stromal cells, while this expression was detected at both positions in canine corneal stromal cells [28]. Moreover, Pax6 was investigated for its crucial role in self-renewal and its ability to sustain holoclone formation of limbal epithelial cells [38].

Moreover, CSSCs were accepted to be natural progenitors of corneal keratocytes from the neural crest [39]. Their ability to differentiate into keratocyte was thus verified by their morphology changing to that of keratocytes, that is, stellate formation with numerous cell processes [40]. The downregulation of proliferation (*Ki67*) and adult stemness-related markers (*Nes*, *CD90*, *Pax6*) and the upregulation of genes associated with keratocytes (*Aldh3a1*, *Aqp1*, *Kera*, and *Lum*) were similar to the study of Kafarnik et al., as it relates to immunofluorescent staining [28].

Since canine cornea donors and grafts are not only inadequate but also have limitations, bioengineered canine cornea is a fascinating alternative. Natural materials have been used to fabricate three-dimensional (3-D) biocompatible scaffolds, including collagen, silk fibroin (SF), and gelatin. Collagen has been widely accepted as a corneal scaffold [41, 42] as well as SF [18, 19, 43, 44]. However, SF shows superior mechanical toughness and slow degradation, which are appropriate for corneal scaffold in high-activity animals [14]. As a result, in this study, SF/G was chosen and successfully fabricated to produce a smooth surface film that acted as a basement membrane of the epithelial layer and a sponge-like scaffold processed abundant ECM component lattice served for corneal stroma. A 130.71 ± 37.12 μm pore size was created using the freeze-dry method, which is consistent with the range of 95.9–150.5 μm that promotes cell attachment and viability [45]. The transparency of scaffolds could be qualitatively observed by placing the scaffold over colored paper. However, the qualitative degree of transparency, the assessment by spectrophotometer in different wavelengths, was not analyzed in this study [46]. Corneal transparency and structure maintenance are correlated with suitable water content. The results of water content indicated this was in the 75–90% range at all time points, which is close to that of normal human cornea [47]. Nevertheless, SF/G corneal film showed a slightly decreased % of water content because of the large amount of non-crosslink

gelatin (70%) and inadequate β-sheet formation from water annealing at RT. Hence, 95˚C water annealing for generating β-sheet up to 60% might be suggested [25]. The degradation rate was evaluated by protease XIV solution as a result of the most efficient enzymatic degradation compared to collagenase IA and α-chymotrypsin in silk protein [48]. SF/G stromal scaffold presented a faster degradation rate than SF/G corneal film, and the result interested in the porosity of scaffold suggested to elevate degradation rate [49]. With regard to mechanical properties, their elastic moduli were in the range of natural human cornea (0.3-7Mpa) [50]. When comparing the two structures, elastic modulus, UTS, and elongation at break of SF/G corneal film were detected as being higher due to the absence of porosity [49]. Concerning this, there have been a few studies on mechanical properties of canine cornea. He et al. demonstrated the mean secant modulus and tangent modulus with equivalent loading at 1.05 ± 0.40 and 1.28 ± 0.47 Mpa, respectively. The mean secant modulus increased from 1.07 ± 0.48 MPa at 1% strain to 2.01 ± 0.98 MPa at 5% strain, while the mean tangent modulus increased from 1.28 ± 0.69 to 3.16 ± 0.71 MPa [51].

Interestingly, Wu et al. 2014 revealed the crucial role of RGD, arginine–glycine–aspartic acid for integrin-related cell attachment [52]. SF itself shows a lack of the RDG sequence. Mixing with gelatin-containing RGD can promote cell adhesion capacity; therefore, an RGD-modified process that would result in chemical residues was unnecessary. In addition, incorporation with gelatin was useful to adjust the suitable degradation rate [23]. In term of mixing ratio, SF/G (a ratio of 30/70 and final solid concentration of 5% w/w) demonstrated the most cell adhesion capacity compared to 0/100, 50/50, 70/30, and 100/0 in the pilot study.

cLESCs seeded SF/G corneal film exhibited a proliferative phase after day 3, followed by plateau phase after day 5. Similar to hLESCs seeded collagen I substrate, the pattern of the growth curve showed a proliferative phase between 3 days and 7 days [53], while the proliferation rate of cCSSCs manifested a plateau phase up to 7 days, followed by a log phase. This scenario can be interpreted as the effect of serum-free media associated with slow cell growth [54]. In contrast, Foster et al. discussed the growth rate of human corneal stromal cells in serum-free media with low glucose after 3 days, which plateaued after 7 days; however, this study was undertaken in a 2-D environment [55].

For cLESCs seeded SF/G corneal film, the expressions of *Rex1*, *Oct4*, and *Abcg2* might have been affected by suitable matrix stiffness of SF/G and cLESCs behavior, as the study of Gouveia et al. described the relationship between soft substrates and LESC markers [56]. The results showed this approach's superior efficiency to maintain ABCG2 and P63 in soft substrate via mechanotransduction [56]. However, the lower expression of *Ki67* at day 28 was correlated with the proliferation assay that exhibited slow proliferation at the late period, which was possibly caused by contact inhibition [57]. Morita et al. and Nam et al. indicated that ABCG2 and P63 are markers of limbal corneal epithelial stem cells (LESCs) and corneal epithelial cell proliferation, respectively; therefore, those makers were validated in this study [27, 58]. P63 can detect both TACs (transient amplifying cells) and stem cells, whereas ABCG2 detects only stem cells. This result suggested that from day 14 to day 28, SF/G corneal film could enhance the differentiation of stem cells into TACs and finally to terminal differentiated cells (corneal epithelial cells) as defined as a greater expression of *Krt3* and *Krt12* [59].

*Rex 1*, *Oct4*, *Pax6*, *Ngfr*, and *Nes* expressions of cCSSCs seeded SF/G stromal scaffold at day 14 were upregulated even in KDM. These outstanding results pointed to a preserved stemness ability of cells seeded in the scaffold, similar to the result of mesenchymal stem cells (MSCs) from juvenile bovine bone marrow that maintain stemness markers in poly (ε-caprolactone) (PCL) nanofibrous scaffolds compared to TCP [60]. In contrast, the adverse results of *Rex1*, *Oct4*, *Pax6* at day 28 alluded to long-term culture and potential of KDM for differentiated keratocytes. In addition, such MSCs also explain the lower expression of *Ki67* in this study since

MSCs are activated toward a fibrotic differentiation pathway in TCP and regulated by an increase in MSC contractility and YAP activation [60]. *CD90* (a well-accepted MSC marker) was downregulated at day 14 and day 28 compared to TCP. As a consequence of CD90 investigations of fibroblasts, cCSSCs seeded SF/G stromal scaffold was commendable in terms of avoiding fibroblast and myofibroblast differentiation, described by downregulation of *CD90* and *ACTA2* [61]. The results supported the advantage of these corneal stromal equivalents, considering fibroblasts and myofibroblasts are responsible for generating unorganized stromal ECM, resulting in opaque corneal scar [62]. In addition, keratocan and lumican, ECM molecules associated with spacing and collagen fibril organization related to corneal transparency, were significantly upregulated at day 28 [63]. Surprisingly, aquaporin-1 (*Aqp1*) water channels, present in human corneal keratocytes associated with cell migration and corneal wounding, were first discovered in canine differentiated keratocytes [64]. Furthermore, *ALDH3A1*, corneal crystallins, plays an essential role in protecting ocular tissue from ultraviolet radiation (UVR)-induced oxidative damage via non-catalytic and catalytic mechanisms [65]. In human studies, *AQP1 and ALDH3A1* were upregulated within 21 days after differentiation. Likewise, in this study, the expressions of *Aqp1* and *Aldh3a1* were significantly upregulated at day 14 and 28, respectively [66].

Bio-fabricated cornea was composed of cLESCs seeded SF/G upon cCSSCs seeded SF/G stromal scaffold and connected by tissue glue sealant (Tisseel®; Baxter Corporation, USA). Fig 6B describes the structural stability corresponding to the ability of the cornea to be manipulatable, transferable, and suturable. The slightly hazy appearance of bio-fabricated cornea highlighted its limitation compared to transparent corneal grafts. Nevertheless, the scaffold is expected to degrade after *in vivo* transplantation and the organization of ECM by differentiated keratocytes, which will result in corneal transparency. Similar to an auto-tissue-engineered lamellar cornea (ATELC), acellular porcine corneal stroma with autologous corneal limbal explants in rabbits showed desirable transparency 20 days after transplantation [67].

A fibrin sealant product, Tisseel (Tisseel®; Baxter Corporation), completely performed a fibrin polymerization to seal and stabilize 2 materials. Fibrinogen, supplied by product, is converted to the fibrin monomer and forms a fibrin polymer which binds the separated tissue and creates hemostasis. Moreover, Tisseel (Tisseel®; Baxter Corporation) has been favored in several oculoplastics surgeries, including sutureless lamellar keratoplasty, pterygium surgery, and management of bleb leaks in humans [68].

cLESCs seeded SF/G corneal film presented successful adherence, proliferation, and stratification of cLESCs upon SF/G. Air-liquid interface cell culture is rationally accepted to promote epithelial proliferation, differentiation, and stratification affected by the shift of oxidative metabolism from the growth phase to the differentiation phase and contributes a polarization effect allowing stratification [69]. In an air-liquid interface environment, H&E represented cell proliferation and 2–3 layers of stratification of cLESCs seeded SF/G corneal film as well as bio-fabricated cornea. Thus, from a logical postulate, cLESCs seeded SF/G corneal film at day 28 obviously presented features of morphological differentiation compared with day 14, characterized by large size, large nucleus, and low nucleo-cytoplasmic ratio [70]. Moreover, strong positive P63 expression was locally detected at the basal layer of the cross-section of the cLESCs seeded SF/G corneal films in the bio-fabricated cornea, where corneal stem cells are located [71]. The space between cLESCs seeded SF/G corneal films and cCSSCs seeded SF/G corneal scaffold in H&E staining appeared 8.5 times bigger than in SEM. Tissue processing during H&E sectioning process might be the cause of this because it was not operated in SEM protocols.

cCSSCs, distributed throughout SF/G stromal scaffold and bio-fabricated cornea at day 14 and 28, exhibited Aldh3a1 and lumican expression, characteristic of the first achievement of

canine keratocyte differentiation in SF/G stromal scaffold. Lumican, a major keratan sulfate proteoglycan responding to corneal transparence, had stronger expression over culturing time related to *Lum* gene expression that upregulated over 1,000 folds. Meanwhile, Aldh3a1 expression was similarly detected on both days [72].

Corneal stroma is an abundant collagenous connective tissue produced by keratocytes. Collagen is responsible for optical transparency, refraction, and mechanical strength [73]. Wu et al. explored the positive relationship between collagen-producing human corneal stromal stem cells with RGD-modified silk scaffold. With RGD coupling, collagen fibrils were robustly aligned and distributed [52]. Additionally, FGF-2 and TGF-β3 favorably activated the production of a stromal-like tissue composed of multilayered lamellae with orthogonally oriented collagen fibrils and the cornea-specific proteins and proteoglycans [74]. The collagen staining result signified that the direction of the collagen could be manipulated via the surface of the scaffold. Nevertheless, SF/G scaffold created an irregular ultrastructure that was difficult to succeed the biomimetic corneal stroma. To approach multilayered lamellae with orthogonally oriented collagen fibrils, topographical modification of scaffold would be beneficial to regulate the collagen's direction. For example, a groove topography successfully manifested well-defined lamella collagen orientation and promoted proper ECM including keratan sulfate, lumican, and keratocan [52].

However, this study produced an insufficient corneal endothelial layer on the posterior side. Corneal endothelium is important to maintain corneal transparency by transporting water out of the stroma, controlling corneal hydration and nutrition [75]. Accordingly, corneal epithelial and stromal equivalents could not be manipulated for penetrating or endothelial keratoplasty. To satisfy many clinical applications, 3 cellular layers should be generated in further experiments.

Regardless, the summary of all aforementioned results contributed to the knowledge of cLESCs and cCSSCs for corneal tissue engineering with SF/G-based scaffold. SF/G would hypothetically be degraded after transplantation, while cLESCs and differentiated keratocytes would be colonized and contribute essential ECM as a native cornea.

## Conclusion

SF/G corneal film and stromal scaffold were achieved to support cell adhesion, viability, and proliferation as well as to promote the differentiation of cLESCs and cCSSCs into keratocytes. Endogenous ECM production exhibited the capability to imitate native cornea after 14 days. This study endeavored to generate stem-cell-incorporated corneal epithelial and stromal equivalents for canine corneal regeneration.

## Supporting information

**S1 Table. General information of the subjects.** Breed, gender and age of the subjects. (DOCX)

## Acknowledgments

The authors thank Assoc.Prof.Dr. Sayamon Srisuwattanasagul and Mr. Kittipot Kongsonthana, Department of Veterinary Anatomy, Faculty of Veterinary Science, Chulalongkorn University, histological and immunocytochemistry analyses; Small Animal Hospital Faculty of Veterinary Science, Rak Na Chan Veterinary Hospital, Phyathai 7 veterinary hospital, Pranee veterinarian Eyes Animal Disease Center for providing eye samples; Professor Kaywalee Chatdarong (DVM, MSc, PhD, DTBT), Department of Obstetrics, Gynaecology and Reproduction,

Faculty of Veterinary Science, Chulalongkorn University, for providing support on qPCR analysis.

## Author Contributions

**Conceptualization:** Watchareewan Rodprasert, Sirirat Nantavisai, Chenphop Sawangmake.

**Data curation:** Chutirat Torsahakul.

**Formal analysis:** Chutirat Torsahakul.

**Funding acquisition:** Chenphop Sawangmake.

**Investigation:** Chutirat Torsahakul.

**Methodology:** Chutirat Torsahakul, Juthamas Ratanavaraporn.

**Project administration:** Chenphop Sawangmake.

**Resources:** Juthamas Ratanavaraporn, Chenphop Sawangmake.

**Supervision:** Nipan Israsena, Supaporn Khramchantuk, Juthamas Ratanavaraporn, Sirakarnt Dhitavat.

**Validation:** Chutirat Torsahakul, Sirirat Nantavisai, Chenphop Sawangmake.

**Visualization:** Chutirat Torsahakul.

**Writing – original draft:** Chutirat Torsahakul.

**Writing – review & editing:** Watchareewan Rodprasert, Sirirat Nantavisai, Chenphop Sawangmake.

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
