## [Decision Letter · Decision Letter 0]

24 Nov 2021

PONE-D-21-30098Bio-fabrication of stem cell-incorporated corneal equivalents from silk fibroin and gelatin-based biomaterial for canine corneal regenerationPLOS ONE

Dear Dr. Sawangmake,

Thank you for submitting your manuscript to PLOS ONE. After careful consideration, we feel that it has merit but does not fully meet PLOS ONE’s publication criteria as it currently stands. Therefore, we invite you to submit a revised version of the manuscript that addresses the points raised during the review process.

We look forward to receiving your revised manuscript.

Kind regards,

Panayiotis Maghsoudlou

Academic Editor

PLOS ONE

Journal Requirements:

Whilst you may use any professional scientific editing service of your choice, PLOS has partnered with both American Journal Experts (AJE) and Editage to provide discounted services to PLOS authors. Both organizations have experience helping authors meet PLOS guidelines and can provide language editing, translation, manuscript formatting, and figure formatting to ensure your manuscript meets our submission guidelines. To take advantage of our partnership with AJE, visit the AJE website (http://aje.com/go/plos) for a 15% discount off AJE services. To take advantage of our partnership with Editage, visit the Editage website (www.editage.com) and enter referral code PLOSEDIT for a 15% discount off Editage services.  If the PLOS editorial team finds any language issues in text that either AJE or Editage has edited, the service provider will re-edit the text for free.

Reviewers' comments:

Reviewer's Responses to Questions

**Comments to the Author**

1. Is the manuscript technically sound, and do the data support the conclusions?

Reviewer #1: Yes

Reviewer #2: Yes

Reviewer #3: Yes

2. Has the statistical analysis been performed appropriately and rigorously? 

Reviewer #1: Yes

Reviewer #2: Yes

Reviewer #3: Yes

3. Have the authors made all data underlying the findings in their manuscript fully available?

Reviewer #1: Yes

Reviewer #2: Yes

Reviewer #3: Yes

4. Is the manuscript presented in an intelligible fashion and written in standard English?

Reviewer #1: No

Reviewer #2: No

Reviewer #3: Yes

5. Review Comments to the Author

Reviewer #1: he authors should check their manuscript for consistency, for instance ml and mL are both used. Also, numbers are written in full and as numbers; please write numbers until twenty in full and higer in numbers.

Language remarks:

45: I have been told never to use abbreviations in the abstract of an article. Even more, they are not defined upon their first use in the abstract.

76: corneal grafts

83-84: the use of the ex. Abbreviation is not common. Please use alternatives such as: for example, e.g. etc. (also on line 91)

92: transparency

107: promising: remove

109: “were replenished the knowledge” � change

143: remove a before 2 ml

302: In brief: … � change the structure of the sentence.

345: twice of… use correct English formulation

361: shield instead of shied

415: incorrect use of language

439 decreased instead of deceased

530 Pax6 was similar pattern is no correct English.

645: promote instead of promotes

647: revise sentence

720: gelatin containing RGD

740: therefor … revise sentence

755: arise from… what is meant here?

Reviewer #2: Bio-fabrication of stem cell-incorporated corneal equivalents from silk fibroin and gelatin-based biomaterial for canine corneal regeneration

The authors propose a new technology for bio-engineering corneal tissue using silk fibroin/gelatin films and scaffolds as careers to grow, respectively, limbal stem cells and corneal stromal stem cells. LSC seeded films and CSSC seeded scaffolds were assembled using tissue glue. Canine stem cells were used to prepare artificial corneas aimed at treating dogs with corneal blindness.

Abstract. Many abbreviations are used in the abstract (LSC, CSSC..). They should be first shown in full (limbal stem cell…). Otherwise the abstract is difficult to read.

The manuscript should be corrected for language by a native English speaker as many sentences are awkward or difficult to understand.

“In humans and animals, corneas consist of 5 recognized layers including 3 cellular layers (epithelium, stroma, and endothelium) and 2 acellular layers (Bowman’s layer and Descemet’s membrane).”

Descemet’s membrane is not a corneal layer. It is the corneal endothelium basement membrane.

“LESCs and CSSCs have been progressed in stem cell properties ex. clonal growth in vitro, extended lifespan, and the ability of differentiation in particular keratocytes.”

What does this sentence mean?

“Various materials have been used to fabricate three-dimensional (3-D) biocompatible scaffolds such as silk protein.”

The main material used as a scaffold is collagen (see Fagerholm, P., Lagali, N.S., Merrett, K. et al., 2010. A biosynthetic alternative to human donor tissue for inducing corneal regeneration: 24-month follow-up of a Phase 1 clinical study. Sci. Transl. Med. Tidu, A., Ghoubay-Benallaoua, D., Teulon, C., et al., 2018. Highly concentrated collagen solutions leading to transparent scaffolds of controlled three-dimensional organizations for corneal epithelial cell colonization. Biomater. Sci…).

Preparation of gelatin. The authors should provide the characteristics of the gelatin used in their study (specie, preparation process, chemical characteristics).

“Unilateral/Bilateral corneas were obtained from fifteen cadaveric healthy dog eyes”

Please provide the spicy.

Figure legends are lacking.

Fig. 1. Images of immunofluorescence of the various markers used to characterize stem cells would be needed. Although informative, RT-PCR graphs do not allow protein staining intensity and localization to be observed.

Fig 1B shows a colony of spindle-shaped fibroblasts that are not features of CSSC. Did the authors obtain spheres when growing CSSC? (see Ghoubay-Benallaoua D, et al. Easy xeno-free and feeder-free method for isolating and growing limbal stromal and epithelial stem cells of the human cornea. PLoS One 2017).

3D colonization of the CSSC seeded scaffolds is an important issue. Fig. 3 does not provide evidence of 3D colonization as SEM is only a surface technique. Please provide cross-sections with the top and bottom surfaces of the scaffold identified.

“Cross-sectional figure demonstrated the upper part of cLESCs seeded SF/G corneal film adhered to the lower part of cCSSCs seeded SF/G stromal scaffold, besides the interconnected space was absent.” This sentence is not supported by Fi. 6 which shows a large space between the film and the scaffold (HES).

FiG. 6 shows the stromal ultrastructure is quite different from the normal corneal stromal ultrastructure. The obtained ultrastructure is not likely to be associated with a high level of transparency as it is completely disorganized.

“By our established technique, the stem cell-incorporated corneal equivalents for canine corneal regeneration were successfully generated.”

This sentence is not supported by the data shown.

Discussion should mention three major study limitations. First, the normal corneal stromal ultrastructure was not regenerated preventing normal transparency to be reached. Second, the material transparency at various light wavelengths was not assessed. Last no attempt to seed the reconstructed tissue with endothelial cells was made. As a result, the resulting bio-engineered tissue cannot be used for penetrating or endothelial keratoplasty.

Reviewer #3: The manuscript titled, "Bio-fabrication of stem cell-incorporated corneal equivalents from silk fibroin and gelatin-based biomaterial for canine corneal regeneration" describes silk fibroin and gelatin mixed biomaterials as a scaffold material for canine corneal tissue regeneration. Authors prepared silk fibroin and primary limbal epithelial stem cells and corneal stromal stem cells for this study. Additionally, cLESCs and cCSSCs cultured SF/G scaffolds were cultured for up to 28 days and analysed for corneal cell specific markers. The corneal regeneration study is not so many until now and this manuscript is accountable for the cell proliferation and tissue regeneration trial.

However, there are points to be addressed prior to publication.

Major points;

1. Discuss about the advantages of gelatin mixing with the silk fibroin. There looks minimal advantage with the addition of gelatin, as it is soluble to water and the gelatin content in the silk fibroin scaffold would be lower than initial mixing ratio.

2. In the title, the expression "Corneal equivalents" is inappropriate because this study does not include corneal endothelium. Corneal endothelium is essential for maintaining transparent cornea by pumping out water and dehydrate corneal stroma.

3. Add discussion about the corneal endothelium for corneal equivalent biofabrication in the discussion part.

4. Figure 7. There looks negligible cell number and collagen amount change during day 14 and day 28. It is recommended to add images of culture day 1 and add quantitative collagen amount data for clear collgen type 1 change with cell culture period.

5. line 457 There are only the mechanical properties of the scaffolds. Recommended to add data for canine corneal tissue mechanical properties. Further, add discussion about the mechanical property data of the scaffolds with the native corneal tissues.

6. Overall, quantitative result data were not written or included in the results parts. The numbers and statistical significance should be included in the manuscript.

Minor points;

1. Use subscript for W0, Wt.

2. Use symbol instead of alphabet "X" in the equation.

3. Add scale bars in the figure legends.

4. line 127 molar concentration is missing. : 9.3M LiBr solution

5. line 154 Detailed description is needed how the air interphase area of scaffolds was removed.

6. line 155 Please describe in detail how the scaffold was made into 1 mm thickness(include tool, procedure and so on).

6. PLOS authors have the option to publish the peer review history of their article (what does this mean?). If published, this will include your full peer review and any attached files.

Reviewer #1: **Yes: **Bert Van den Bogerd

Reviewer #2: No

Reviewer #3: No

---

## [Author Response · Author response to Decision Letter 0]

7 Jan 2022

I'm satisfied all of the reviewers' comments.

---

## [Editor Report · Decision Letter 1]

13 Jan 2022

Bio-fabrication of stem-cell-incorporated corneal epithelial and stromal equivalents from silk fibroin and gelatine-based biomaterial for canine corneal regeneration

PONE-D-21-30098R1

Dear Dr. Sawangmake,

We’re pleased to inform you that your manuscript has been judged scientifically suitable for publication and will be formally accepted for publication once it meets all outstanding technical requirements.

Kind regards,

Panayiotis Maghsoudlou

Academic Editor

PLOS ONE

---

## [Editor Report · Acceptance letter]

27 Jan 2022

PONE-D-21-30098R1 

Bio-fabrication of stem-cell-incorporated corneal epithelial and stromal equivalents from silk fibroin and gelatin-based biomaterial for canine corneal regeneration

Dear Dr. Sawangmake:

I'm pleased to inform you that your manuscript has been deemed suitable for publication in PLOS ONE. Congratulations! Your manuscript is now with our production department. 

Kind regards, 

on behalf of

Dr. Panayiotis Maghsoudlou 

Academic Editor

PLOS ONE